# Deep Structural Causal Models
# for Tractable Counterfactual Inference

**Nick Pawlowski**[*]
Imperial College London
np716@imperial.ac.uk

**Daniel C. Castro**[*]
Imperial College London
dc315@imperial.ac.uk

**Ben Glocker**
Imperial College London
b.glocker@imperial.ac.uk

## Abstract

We formulate a general framework for building structural causal models (SCMs) with deep learning components. The proposed approach employs normalising flows and variational inference to enable tractable inference of exogenous noise variables—a crucial step for counterfactual inference that is missing from existing deep causal learning methods. Our framework is validated on a synthetic dataset built on MNIST as well as on a real-world medical dataset of brain MRI scans. Our experimental results indicate that we can successfully train deep SCMs that are capable of all three levels of Pearl's ladder of causation: association, intervention, and counterfactuals, giving rise to a powerful new approach for answering causal questions in imaging applications and beyond. The code for all our experiments is available at https://github.com/biomedia-mira/deepscm.

## 1 Introduction

Many questions in everyday life as well as in scientific inquiry are causal in nature: "How would the climate have changed if we'd had less emissions in the '80s?", "How fast could I run if I hadn't been smoking?", or "Will my headache be gone if I take that pill?". None of those questions can be answered with statistical tools alone, but require methods from causality to analyse interactions with our environment (interventions) and hypothetical alternate worlds (counterfactuals), going beyond joint, marginal, and conditional probabilities [1]. Even though these are natural lines of reasoning, their mathematical formalisation under a unified theory is relatively recent [2].

In some statistics-based research fields, such as econometrics or epidemiology, the use of causal inference methods has been established for some time [3, 4]. However, causal approaches have been introduced into deep learning (DL) only very recently [5]. For example, research has studied the use of causality for disentanglement [6, 7], causal discovery [8, 9], and for deriving causality-inspired explanations [10, 11] or data augmentations [12]. Causal DL models could be capable of learning relationships from complex high-dimensional data and of providing answers to interventional and counterfactual questions, although current work on deep counterfactuals is limited by modelling only direct cause-effect relationships [11] or instrumental-variable scenarios [13], or by not providing a full recipe for tractable counterfactual inference [14].

The integration of causality into DL research promises to enable novel scientific advances as well as to tackle known shortcomings of DL methods: DL is known to be susceptible to learning spurious correlations and amplifying biases [e.g. 15], and to be exceptionally vulnerable to changes in the input distribution [16]. By explicitly modelling causal relationships and acknowledging the difference between causation and correlation, causality becomes a natural field of study for improving the transparency, fairness, and robustness of DL-based systems [17, 18]. Further, the tractable inference of deep counterfactuals enables novel research avenues that aim to study causal reasoning on a

---

[*]Joint first authors.

per-instance rather than population level, which could lead to advances in personalised medicine as well as in decision-support systems, more generally.

In this context, our work studies the use of DL-based causal mechanisms and establishes effective ways of performing counterfactual inference with fully specified causal models with no unobserved confounding. Our main contributions are: 1) a unified framework for structural causal models using modular deep mechanisms; 2) an efficient approach to estimating counterfactuals by inferring exogenous noise via variational inference or normalising flows; 3) case studies exemplifying how to apply deep structural causal models and perform counterfactual inference. The paper is organised as follows: we first review structural causal models and discuss how to leverage deep mechanisms and enable tractable counterfactual inference. Second, we compare our work to recent progress in deep causal learning in light of Pearl's ladder of causation [19]. Finally, we apply deep structural causal models to a synthetic experiment as well as to modelling brain MRI scans, demonstrating the practical utility of our framework in answering counterfactual questions.

## 2 Deep Structural Causal Models

We consider the problem of modelling a collection of $K$ random variables $\mathbf{x} = (x_1, \ldots, x_K)$. By considering causal relationships between them, we aim to build a model that not only is capable of generating convincing novel samples, but also satisfies all three rungs of the causation ladder [19]. The first level, **association**, describes reasoning about passively observed data. This level deals with correlations in the data and questions of the type *"What are the odds that I observe. . . ?"*, which relates purely to marginal, joint, and conditional probabilities. **Intervention** concerns interactions with the environment. It requires knowledge beyond just observations, as it relies on structural assumptions about the underlying data-generating process. Characteristic questions ask about the effects of certain actions: *"What happens if I do. . . ?"*. Lastly, **counterfactuals** deal with retrospective hypothetical scenarios. Counterfactual queries leverage functional models of the generative processes to imagine alternative outcomes for individual data points, answering *"What if I had done A instead of B?"*. Arguably, such questions are at the heart of scientific reasoning (and beyond), yet are less well-studied in the field of machine learning. The three levels of causation can be operationalised by employing structural causal models (SCMs)[2], recapitulated in the next section.

### 2.1 Background on structural causal models

A structural causal model $\mathfrak{G} := (\mathbf{S}, P(\boldsymbol{\epsilon}))$ consists of a collection $\mathbf{S} = (f_1, \ldots, f_K)$ of structural assignments $x_k := f_k(\epsilon_k; \mathbf{pa}_k)$ (called *mechanisms*), where $\mathbf{pa}_k$ is the set of parents of $x_k$ (its *direct causes*), and a joint distribution $P(\boldsymbol{\epsilon}) = \prod_{k=1}^{K} P(\epsilon_k)$ over mutually independent exogenous noise variables (i.e. unaccounted sources of variation). As assignments are assumed acyclic, relationships can be represented by a directed acyclic graph (DAG) with edges pointing from causes to effects, called the *causal graph* induced by $\mathfrak{G}$. Every SCM $\mathfrak{G}$ entails a unique joint observational distribution $P_{\mathfrak{G}}(\mathbf{x})$, satisfying the causal Markov assumption: each variable is independent of its non-effects given its direct causes. It therefore factorises as $P_{\mathfrak{G}}(\mathbf{x}) = \prod_{k=1}^{K} P_{\mathfrak{G}}(x_k | \mathbf{pa}_k)$, where each conditional distribution $P_{\mathfrak{G}}(x_k | \mathbf{pa}_k)$ is determined by the corresponding mechanism and noise distribution [1].

Crucially, unlike conventional Bayesian networks, the conditional factors above are imbued with a causal interpretation. This enables $\mathfrak{G}$ to be used to predict the effects of *interventions*, defined as substituting one or multiple of its structural assignments, written as 'do($\cdots$)'. In particular, a constant reassignment of the form do($x_k := a$) is called an atomic intervention, which disconnects $x_k$ from all its parents and represents a direct manipulation disregarding its natural causes.

While the observational distribution relates to statistical associations and interventions can predict causal effects, SCMs further enable reasoning about *counterfactuals*. In contrast to interventions, which operate at the population level—providing aggregate statistics about the effects of actions (i.e. noise sampled from the prior, $P(\boldsymbol{\epsilon})$)—a counterfactual is a query at the unit level, where the structural assignments ('mechanisms') are changed but the exogenous noise is identical to that of the observed datum ($P(\boldsymbol{\epsilon} | \mathbf{x})$) [1, 2]. These are hypothetical retrospective interventions (cf. potential outcome), given an observed outcome: 'What would $x_i$ have been if $x_j$ were different, given that we observed $\mathbf{x}$?'. This type of question effectively offers explanations of the data, since we can analyse

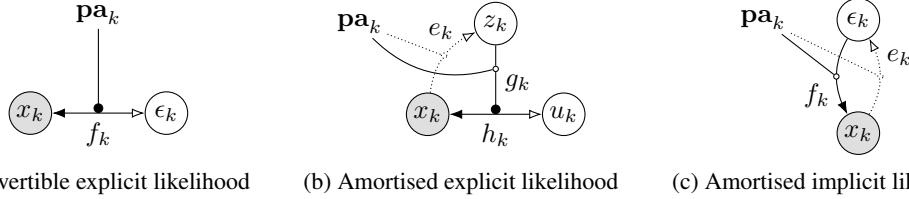

(a) Invertible explicit likelihood     (b) Amortised explicit likelihood     (c) Amortised implicit likelihood

Figure 1: Classes of deep causal mechanisms considered in this work. Bi-directional arrows indicate invertible transformations, optionally conditioned on other inputs (edges ending in black circles). Black and white arrowheads refer resp. to the generative and abductive directions, while dotted arrows depict an amortised variational approximation. Here, $f_k$ is the forward model, $e_k$ is an encoder that amortises abduction in non-invertible mechanisms, $g_k$ is a 'high-level' non-invertible branch (e.g. a probabilistic decoder), and $h_k$ is a 'low-level' invertible mapping (e.g. reparametrisation).

the changes resulting from manipulating each variable. Counterfactual queries can be mathematically formulated as a three-step procedure [2, Ch. 7]:

1. **Abduction:** Predict the 'state of the world' (the exogenous noise, $\epsilon$) that is compatible with the observations, $\mathbf{x}$, i.e. infer $P_{\mathfrak{G}}(\epsilon|\mathbf{x})$.

2. **Action:** Perform an intervention (e.g. $\mathrm{do}(x_k := \widetilde{x}_k)$) corresponding to the desired manipulation, resulting in a modified SCM $\widetilde{\mathfrak{G}} = \mathfrak{G}_{\mathbf{x};\mathrm{do}(\widetilde{x}_k)} = (\widetilde{\mathbf{S}}, P_{\mathfrak{G}}(\epsilon|\mathbf{x}))$ [1, Sec. 6.4].

3. **Prediction:** Compute the quantity of interest based on the distribution entailed by the counterfactual SCM, $P_{\widetilde{\mathfrak{G}}}(\mathbf{x})$.

With these operations in mind, the next section explores a few options for building flexible, expressive, and counterfactual-capable functional mechanisms for highly structured data.

## 2.2 Deep mechanisms

In statistical literature (e.g. epidemiology, econometrics, sociology), SCMs are typically employed with simple linear mechanisms (or generalised linear models, involving an output non-linearity). Analysts attach great importance to the regression weights, as under certain conditions these may be readily interpreted as estimates of the causal effects between variables. While this approach generally works well for scalar variables and can be useful for decision-making, it is not flexible enough to model higher-dimensional data such as images. Solutions to this limitation have been proposed by introducing deep-learning techniques into causal inference [8, 14].

We call an SCM that uses deep-learning components to model the structural assignments a *deep structural causal model* (DSCM). In DSCMs, the inference of counterfactual queries becomes more complex due to the potentially intractable abduction step (inferring the posterior noise distribution, as defined above). To overcome this, we propose to use recent advances in normalising flows and variational inference to model mechanisms for composable DSCMs that enable tractable counterfactual inference. While here we focus on continuous data, DSCMs also fully support discrete variables without the need for relaxations (see Appendix C). We consider three types of mechanisms that differ mainly in their invertibility, illustrated in Fig. 1.

**Invertible, explicit:** Normalising flows model complex probability distributions using transformations from simpler base distributions with same dimensionality [20]. For an observed variable $x$, diffeomorphic transformation $f$, and base variable $\epsilon \sim P(\epsilon)$ such that $x = f(\epsilon)$, the output density $p(x)$ can be computed as $p(x) = p(\epsilon)|\det \nabla f(\epsilon)|^{-1}$, evaluated at $\epsilon = f^{-1}(x)$ [21, 22]. For judicious choices of $f$, the Jacobian $\nabla f$ may take special forms with efficiently computable determinant, providing a flexible and tractable probabilistic model whose parameters can be trained via exact maximum likelihood. Furthermore, flows can be made as expressive as needed by composing sequences of simple transformations. For more information on flow-based models, refer to the comprehensive survey by Papamakarios et al. [22]. Note that this class of models also subsumes the typical location-scale and inverse cumulative distribution function transformations used in the reparametrisation trick [23, 24], as well as the Gumbel trick for discrete variable relaxations [25, 26].

Although normalising flows were originally proposed for unconditional distributions, they have been extended to conditional densities [27], including in high dimensions [28, 29], by parametrising the transformation as $x = f(\epsilon; \mathbf{pa}_X)$, assumed invertible in the first argument. In particular, conditional flows can be adopted in DSCMs to represent invertible, explicit-likelihood mechanisms (Fig. 1a):

$$x_i := f_i(\epsilon_i; \mathbf{pa}_i), \qquad p(x_i | \mathbf{pa}_i) = p(\epsilon_i) \cdot |\det \nabla_{\epsilon_i} f_i(\epsilon_i; \mathbf{pa}_i)|^{-1} \big|_{\epsilon_i = f_i^{-1}(x_i; \mathbf{pa}_i)}. \qquad (1)$$

**Amortised, explicit:** Such invertible architectures typically come with heavy computational requirements when modelling high-dimensional observations, because all intermediate operations act in the space of the data. Instead, it is possible to use arbitrary functional forms for the structural assignments, at the cost of losing invertibility and tractable likelihoods $p(x_k | \mathbf{pa}_k)$. Here, we propose to separate the assignment $f_k$ into a 'low-level', invertible component $h_k$ and a 'high-level', non-invertible part $g_k$—with a corresponding noise decomposition $\epsilon_k = (u_k, z_k)$—such that

$$x_k := f_k(\epsilon_k; \mathbf{pa}_k) = h_k(u_k; g_k(z_k; \mathbf{pa}_k), \mathbf{pa}_k), \qquad P(\epsilon_k) = P(u_k)P(z_k). \qquad (2)$$

In such a decomposition, the invertible transformation $h_k$ can be made shallower, while the upstream non-invertible $g_k$ maps from a lower-dimensional space and is expected to capture more of the high-level structure of the data. Indeed, a common implementation of this type of model for images would involve a probabilistic decoder, where $g_k$ may be a convolutional neural network, predicting the parameters of a simple location-scale transformation performed by $h_k$ [24].

As the conditional likelihood $p(x_k | \mathbf{pa}_k)$ in this class of models is no longer tractable because $z_k$ cannot be marginalised out, it may alternatively be trained with amortised variational inference. Specifically, we can introduce a variational distribution $Q(z_k | x_k, \mathbf{pa}_k)$ to formulate a lower bound on the true marginal conditional log-likelihood, which will be maximised instead:

$$\log p(x_k | \mathbf{pa}_k) \geq \mathbb{E}_{Q(z_k | x_k, \mathbf{pa}_k)}[\log p(x_k | z_k, \mathbf{pa}_k)] - D_{\mathrm{KL}}[Q(z_k | x_k, \mathbf{pa}_k) \| P(z_k)]. \qquad (3)$$

The argument of the expectation in this lower bound can be calculated similarly to Eq. (1):

$$p(x_k | z_k, \mathbf{pa}_k) = p(u_k) \cdot |\det \nabla_{u_k} h_k(u_k; g_k(z_k, \mathbf{pa}_k), \mathbf{pa}_k)|^{-1} \big|_{u_k = h_k^{-1}(x_k; g_k(z_k, \mathbf{pa}_k), \mathbf{pa}_k)}. \qquad (4)$$

The approximate posterior distribution $Q(z_k | x_k, \mathbf{pa}_k)$ can for example be realised by an encoder function, $e_k(x_k; \mathbf{pa}_k)$, that outputs the parameters of a simple distribution over $z_k$ (Fig. 1b), as in the auto-encoding variational Bayes (AEVB) framework [24].

**Amortised, implicit:** While the models above rely on (approximate) maximum-likelihood as training objective, it is admissible to train a non-invertible mechanism as a conditional implicit-likelihood model (Fig. 1c), optimising an adversarial objective [30–32]. Specifically, a deterministic encoder $e_j$ would strive to fool a discriminator function attempting to tell apart tuples of encoded real data $(x_j, e_j(x_j; \mathbf{pa}_j), \mathbf{pa}_j)$ and generated samples $(f_j(\epsilon_j; \mathbf{pa}_j), \epsilon_j, \mathbf{pa}_j)$. This class of mechanism is proposed here for completeness, without empirical evaluation. However, following initial dissemination of our work, Dash and Sharma [33] reproduced our Morpho-MNIST experiments (Section 4) and demonstrated these amortised implicit-likelihood mechanisms can achieve comparable performance.

## 2.3 Deep counterfactual inference

Now equipped with effective deep models for representing mechanisms in DSCMs, we discuss the inference procedure allowing us to compute answers to counterfactual questions.

**Abduction:** As presented in Section 2.1, the first step in computing counterfactuals is abduction, i.e. to predict the exogenous noise, $\epsilon$, based on the available evidence, $\mathbf{x}$. Because each noise variable is assumed to affect only the respective observed variable, $(\epsilon_k)_{k=1}^{K}$ are conditionally independent given $\mathbf{x}$, therefore this posterior distribution factorises as $P_{\mathfrak{G}}(\boldsymbol{\epsilon} | \mathbf{x}) = \prod_{k=1}^{K} P_{\mathfrak{G}}(\epsilon_k | x_k, \mathbf{pa}_k)$. In other words, it suffices to infer the noise independently for each mechanism, given the observed values of the variable and of its parents[3].

For invertible mechanisms, the noise variable can be obtained deterministically and exactly by just inverting the mechanism: $\epsilon_i = f_i^{-1}(x_i; \mathbf{pa}_i)$. Similarly, implicit-likelihood mechanisms can be approximately inverted by using the trained encoder function: $\epsilon_j \approx e_j(x_j; \mathbf{pa}_j)$.

Some care must be taken in the case of amortised, explicit-likelihood mechanisms, as the 'high-level' noise $z_k$ and 'low-level' noise $u_k$ are not independent given $x_k$. Recalling that this mechanism is trained along with a conditional probabilistic encoder, $Q(z_k | e_k(x_k; \mathbf{pa}_k))$, the noise posterior can be approximated as follows, where $\delta_w(\,\cdot\,)$ denotes the Dirac delta distribution centred at $w$:

$$
\begin{aligned}
P_{\mathfrak{G}}(\epsilon_k | x_k, \mathbf{pa}_k) &= P_{\mathfrak{G}}(z_k | x_k, \mathbf{pa}_k)\, P_{\mathfrak{G}}(u_k | z_k, x_k, \mathbf{pa}_k) \\
&\approx Q(z_k | e_k(x_k; \mathbf{pa}_k))\, \delta_{h_k^{-1}(x_k; g_k(z_k; \mathbf{pa}_k), \mathbf{pa}_k)}(u_k)\,.
\end{aligned}
\tag{5}
$$

**Action:** The causal graph is then modified according to the desired hypothetical intervention(s), as in the general case (Section 2.1). For each intervened variable $x_k$, its structural assignment is replaced either by a constant, $x_k \coloneqq \widetilde{x}_k$—making it independent of its former parents (direct causes, $\mathbf{pa}_k$) and of its exogenous noise ($\epsilon_k$)—or by a surrogate mechanism $x_k \coloneqq \widetilde{f}_k(\epsilon_k; \widetilde{\mathbf{pa}}_k)$, forming a set of counterfactual assignments, $\widetilde{\mathbf{S}}$. This then defines a counterfactual SCM $\widetilde{\mathfrak{G}} = (\widetilde{\mathbf{S}}, P_{\mathfrak{G}}(\epsilon | \mathbf{x}))$.

**Prediction:** Finally, we can sample from $\widetilde{\mathfrak{G}}$. Noise variables that were deterministically inverted (either exactly or approximately) can simply be plugged back into the respective forward mechanism to determine the new output value. Notice that this step is redundant for observed variables that are not descendants of the ones being intervened upon, as they will be unaffected by the changes.

As mentioned above, the posterior distribution over $(z_k, u_k)$ for an amortised, explicit-likelihood mechanism does not factorise (Eq. (5)), and the resulting distribution over the counterfactual $x_k$) cannot be characterised explicitly. However, sampling from it is straightforward, such that we can approximate the counterfactual distribution via Monte Carlo as follows, for each sample $s$:

$$
\begin{aligned}
z_k^{(s)} &\sim Q(z_k | e_k(x_k; \mathbf{pa}_k)) \\
u_k^{(s)} &= h_k^{-1}(x_k; g_k(z_k^{(s)}; \mathbf{pa}_k), \mathbf{pa}_k) \\
\widetilde{x}_k^{(s)} &= \widetilde{h}_k(u_k^{(s)}; \widetilde{g}_k(z_k^{(s)}; \widetilde{\mathbf{pa}}_k), \widetilde{\mathbf{pa}}_k)\,.
\end{aligned}
\tag{6}
$$

Consider an uncorrelated Gaussian decoder for images as a concrete example, predicting vectors of means and variances for each pixel of $x_k$: $g_k(z_k; \mathbf{pa}_k) = (\mu(z_k; \mathbf{pa}_k), \sigma^2(z_k; \mathbf{pa}_k))$, with the low-level reparametrisation given by $h_k(u_k; (\mu, \sigma^2), \mathbf{pa}_k) = \mu + \sigma^2 \odot u_k$. Exploiting the reparametrisation trick, counterfactuals that preserve $x_k$'s mechanism can be computed simply as

$$
u_k^{(s)} = (x_k - \mu(z_k^{(s)}; \mathbf{pa}_k)) \oslash \sigma(z_k^{(s)}; \mathbf{pa}_k), \qquad \widetilde{x}_k^{(s)} = \mu(z_k^{(s)}; \widetilde{\mathbf{pa}}_k) + \sigma(z_k^{(s)}; \widetilde{\mathbf{pa}}_k) \odot u_k^{(s)}\,,
$$

where $\oslash$ and $\odot$ denote element-wise division and multiplication, respectively. In particular, in the constant-variance setting adopted for our experiments, counterfactuals further simplify to

$$
\widetilde{x}_k^{(s)} = x_k + [\mu(z_k^{(s)}; \widetilde{\mathbf{pa}}_k) - \mu(z_k^{(s)}; \mathbf{pa}_k)]\,.
$$

This showcases how true image counterfactuals are able to retain pixel-level details. Typical conditional generative models would output only $\mu(z_k; \widetilde{\mathbf{pa}}_k)$ (which is often blurry in vanilla variational auto-encoders [34]), or would in addition have to sample $P(u_k)$ (resulting in noisy images).

# 3 Related Work

Deep generative modelling has seen a wide range of contributions since the popularisation of variational auto-encoders (VAEs) [24], generative adversarial networks (GANs) [35], and normalising flows [21]. These models have since been employed to capture conditional distributions [27, 29, 32, 36], and VAEs and GANs were also extended to model structured data by incorporating probabilistic graphical models [37–39]. In addition, deep generative models have been heavily used for (unsupervised) representation learning with an emphasis on disentanglement [40–43]. However, even when these methods faithfully capture the distribution of observed data, they are capable of fulfilling only the association rung of the ladder of causation.

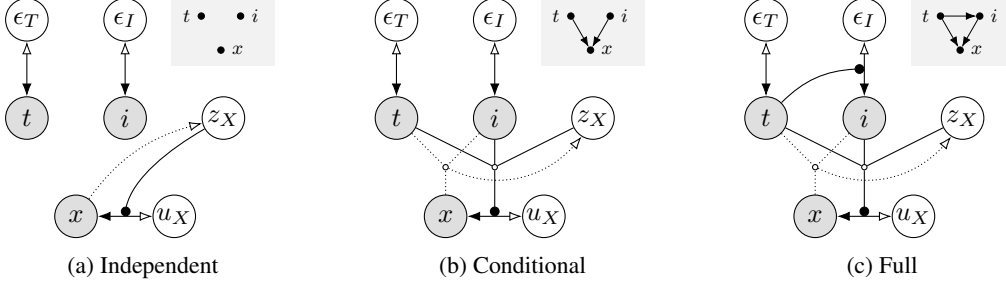

Figure 2: Computational graphs of the structural causal models for the Morpho-MNIST example. The image is denoted by $x$, stroke thickness by $t$, and image intensity by $i$. The corresponding causal diagrams are displayed in the top-right corners.

Interventions build on the associative capabilities of probabilistic models to enable queries related to changes in causal mechanisms. By integrating a causal graph into the connectivity of a deep model, it is possible to perform interventions with GANs [14] and causal generative NNs [8]. VAEs can also express causal links using specific covariance matrices between latent variables, which however restrict the dependences to be linear [6]. Alternatively, assuming specific causal structures, Tran and Blei [44] and Louizos et al. [45] proposed different approaches for estimating causal effects in the presence of unobserved confounders. Despite reaching the second rung of the causal ladder, all of these methods lack tractable abduction capabilities and therefore cannot generate counterfactuals.

Some machine-learning tasks such as explainability, image-to-image translation, or style transfer are closely related to counterfactual queries of the sort 'How would $x$ (have to) change if we (wished to) modify $y$?'. Here, $y$ could be the style of a picture for style transfer [46], the image domain (e.g. drawing to photo) for image-to-image translation [47], the age of a person in natural images [48] or medical scans [49], or a predicted output for explainability [11]. However, these approaches do not explicitly model associations, interventions, nor causal structure. Potentially closest to our work is a method for counterfactual explainability of visual models, which extends CausalGANs [14] to predict reparametrised distributions over image attributes following an assumed causal graph [10]. However, this approach performs no abduction step, instead resampling the noise of attributes downstream from the intervention(s), and does not include a generative model of imaging data. To the best of our knowledge, the proposed DSCM framework is the first flexible approach enabling end-to-end training and tractable inference on all three levels of the ladder of causation for high-dimensional data.

## 4 Case Study 1: Morpho-MNIST

We consider the problem of modelling the causal model of a synthetic dataset based on MNIST digits [50], where we defined stroke thickness to cause the brightness of each digit: thicker digits are brighter whereas thinner digits are dimmer. This simple dataset allows for examining the three levels of causation in a controlled and measurable environment. We use morphological transformations on MNIST [51] to generate a dataset with known causal structure and access to the 'true' process of generating counterfactuals. The SCM designed for this synthetic dataset is as follows:

$$
\begin{aligned}
t &:= f_T^*(\epsilon_T^*) = 0.5 + \epsilon_T^*, & \epsilon_T^* &\sim \Gamma(10, 5), \\
i &:= f_I^*(\epsilon_I^*; t) = 191 \cdot \sigma(0.5 \cdot \epsilon_I^* + 2 \cdot t - 5) + 64, & \epsilon_I^* &\sim \mathcal{N}(0, 1), \quad (7) \\
x &:= f_X^*(\epsilon_X^*; i, t) = \text{SetIntensity}(\text{SetThickness}(\epsilon_X^*; t); i), & \epsilon_X^* &\sim \text{MNIST},
\end{aligned}
$$

where $\text{SetIntensity}(\,\cdot\,; i)$ and $\text{SetThickness}(\,\cdot\,; t)$ refer to the operations that act on an image of a digit and set its intensity to $i$ and thickness to $t$ (see Appendix A.1 for details), $x$ is the resulting image, $\epsilon^*$ is the exogenous noise for each variable, and $\sigma(\,\cdot\,)$ is the logistic sigmoid.

We use this setup to study the capabilities of our framework in comparison to models with less causal structure. We adapt the true causal graph from Eq. (7) and model thickness and intensity using (conditional) normalising flows and employ a conditional VAE for modelling the image. In particular, we adopt the causal graphs shown in Fig. 2 and test a fully independent model (Fig. 2a), a conditional decoder model (Fig. 2b), as well as our full causal model (Fig. 2c). All our experiments

Table 1: Comparison of the associative abilities of the models shown in Fig. 2. The image is denoted by $x$, thickness by $t$, and intensity by $i$. Quantities with $\geq$ are lower bounds. MAE refers to the mean absolute error between pixels of the original image and of its reconstruction.

| Model | $\log p(x,t,i) \geq$ | $\log p(x\|t,i) \geq$ | $\log p(t)$ | $\log p(i\|t)$ | $\mathrm{MAE}(x,x')$ |
|---|---|---|---|---|---|
| Independent | $-5925.26$ | $-5919.14$ | $-0.93$ | $-5.19$ | $4.50$ |
| Conditional | $-5526.50$ | $-5520.37$ | $-0.93$ | $-5.19$ | $4.26$ |
| Full | $-5692.94$ | $-5687.71$ | $-0.93$ | $-4.30$ | $4.43$ |

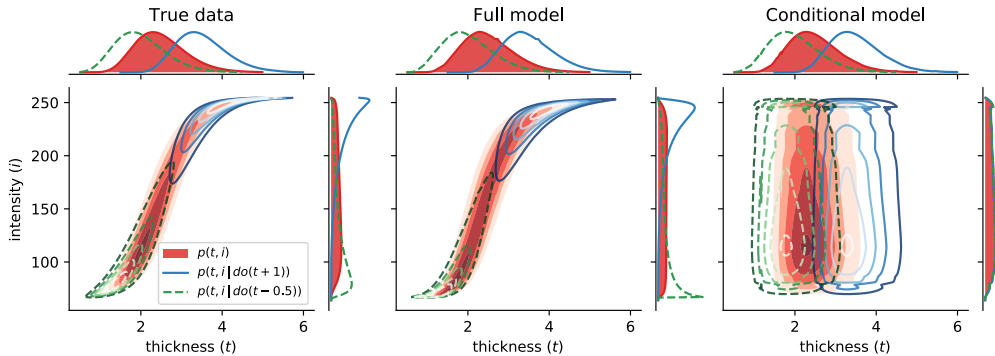

Figure 3: Distributions of thickness and intensity in the true data (left), and learned by the full (centre) and conditional (right) models. Contours depict the observational (red, shaded) and interventional joint densities for $\mathrm{do}(t := f_T(\epsilon_T) + 1)$ (blue, solid) and $\mathrm{do}(t := f_T(\epsilon_T) - 0.5)$ (green, dashed).

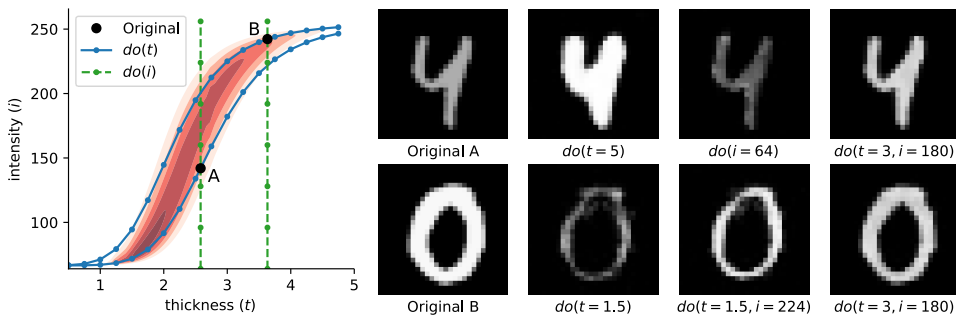

Figure 4: Counterfactuals generated by the full model. (left) Counterfactual 'trajectories' of two original samples, A and B, as their thickness and intensity are modified, overlaid on the learned joint density $p(t,i)$. (right) Original and counterfactual images corresponding to samples A and B.

were implemented within PyTorch [52] using the Pyro probabilistic programming framework [53], and implementation details can be found in Appendices A.2 and B.2.

We quantitatively compare the associative capabilities of all models by evaluating their evidence lower bound (Eq. (3)), log-likelihoods and reconstruction errors as shown in Table 1. We find that performance improves consistently with the model's capabilities: enabling conditional image generation improves $p(x|t,i)$, and adding a causal dependency between $t$ and $i$ improves $p(i|t)$. Further, we examine samples of the conditional and unconditional distributions in Appendix A.3.1.

The interventional distributions can be directly compared to the true generative process. Figure 3 shows that the densities predicted by our full model after intervening on $t$ closely resemble the true behaviour. The conditional and independent models operate equivalently to each other and are incapable of modelling the relationship between $t$ and $i$, capturing only their marginal distributions.

In this special case, where the true data generating process is known, it is possible to evaluate against reference counterfactuals that are impossible to obtain in most real-world scenarios. We compare

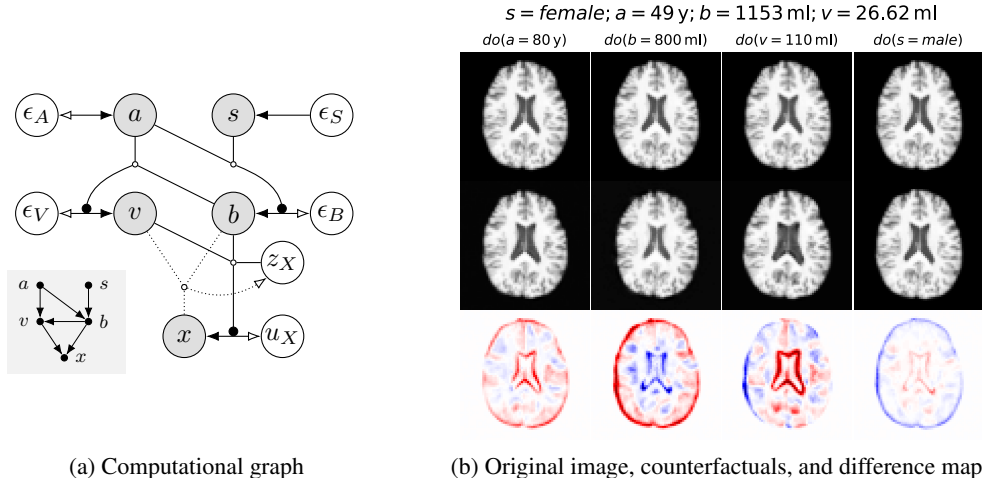

(a) Computational graph      (b) Original image, counterfactuals, and difference maps

Figure 5: Brain imaging example. Variables are image ($x$), age ($a$), sex ($s$), and brain ($b$) and ventricle ($v$) volumes. The counterfactuals show different interventions on the same original brain.

all models on the task of generating counterfactuals with intervention $do(t+2)$ and compute the mean absolute errors between the generated and the reference counterfactual image. For this task, the models perform in order of their complexity: the independent model achieved $41.6$, the conditional $31.8$, and the full model achieved a MAE of $17.6$. This emphasises that, although wrongly specified models will give wrong answers to counterfactual queries (and interventions; see Fig. 3), the results are consistent with the assumptions of each model. The independent model lacks any relationship of the image on thickness and intensity and therefore does not change the image under the given intervention. The conditional model does not model any dependency of intensity on thickness, which leads to counterfactuals with varying thickness but constant intensity. Examples of previously unseen images and generated counterfactuals using the full model are shown in Fig. 4 for qualitative examination. We see that our model is capable of generating convincing counterfactuals that preserve the digit identity while changing thickness and intensity consistently with the underlying true causal model.

## 5   Case Study 2: Brain Imaging

Our real-world application touches upon fundamental scientific questions in the context of medical imaging: how would a person's anatomy change if particular traits were different? We illustrate with a simplified example that our DSCM framework may provide the means to answer such counterfactual queries, which may enable entirely new research into better understanding the physical manifestation of lifestyle, demographics, and disease. Note that any conclusions drawn from a model built in this framework are strictly contingent on the correctness of the assumed SCM. Here, we model the appearance of brain MRI scans given the person's age and biological sex, as well as brain and ventricle volumes[4], using population data from the UK Biobank [54]. Ventricle and total brain volumes are two quantities that are closely related to brain age [55] and can be observed relatively easily. We adopt the causal graph shown in Fig. 5a and otherwise follow the same training procedure as for the MNIST experiments.[5]

The learned DSCM is capable of all three levels of the causal hierarchy. We present the analysis of lower levels in Appendix B.3.1 and focus here on counterfactuals, shown in Fig. 5b (more examples in Appendix B.3.2). The difference maps show plausible counterfactual changes: increasing age causes slightly larger ventricles while decreasing the overall brain volume (first column). In contrast, directly changing brain volume has an opposite effect on the ventricles compared to changing age (second column). Intervening on ventricle volume has a much more localised effect (third column), while

intervening on the categorical variable of biological sex has smaller yet more diffuse effects. Note how the anatomical 'identity' (such as the cortical folding) is well preserved after each intervention.

## 6   Conclusion

We introduce a novel general framework for fitting SCMs with deep mechanisms. Our deep SCM (DSCM) framework fulfils all three rungs of Pearl's causal hierarchy—in particular, it is the first to enable efficient abduction of exogenous noise, permitting principled counterfactual inference. We demonstrate the potential of DSCMs with two case studies: a synthetic task of modelling Morpho-MNIST digits with a known causal structure and a real-world example with brain MRI.

The ability to correctly generate plausible counterfactuals could greatly benefit a wide variety of possible applications, e.g.: *explainability*, where differences between observed and counterfactual data can suggest causal explanations of outcomes; *data augmentation*, as counterfactuals can extrapolate beyond the range of observed data (e.g. novel combinations of attributes); and *domain adaptation*, since including the source of the data as an indicator variable in the causal model could enable generating counterfactual examples in a relevant target domain.

The proposed method does not come without limitations to be investigated in future work. Like the related approaches, the current setup precludes unobserved confounding and requires all variables to be observed when training and computing a counterfactual, which may limit its applicability in certain scenarios. This could be alleviated by imputing the missing data via MCMC or learning auxiliary distributions. Further work should study more closely the training dynamics of deep mechanisms in SCMs: while not observed in our experiments, neural networks may not learn to cleanly disentangle the roles of its inputs on the output as expected. This could call for custom counterfactual regularisation, similar to losses used in image-to-image translation [49] and explainability [11]. The use of such flexible models also raises questions about the identifiability of the 'true' mechanism, as counterfactuals may not be uniquely defined. For real datasets, counterfactual evaluation is possible only in very constrained settings, as generally true counterfactuals can never be observed. For example, assuming our brain graph in Section 5 is correct, we may use a second MRI scan of a brain a few years later as an approximate counterfactual on age. Lastly, it would be interesting to examine whether this framework can be applied to causal discovery, attempting to uncover plausible causal structures from data.

## Broader Impact

Causal inference can be applied to a wide range of applications, promising to provide a deeper understanding of the observed data and prevent the fitting of spurious correlations. Our research presents a methodological contribution to the causal literature proposing a framework that combines causal models and deep learning to facilitate modelling high-dimensional data.

Because of the general applicability of deep learning and causal inference, our framework could have a broad impact of enabling fairer machine learning models explicitly modelling causal mechanisms, reducing spurious correlations and tackling statistical and societal biases. The resulting models offer better interpretability due to counterfactual explanations and could yield novel understanding through causal discovery.

However, causal modelling relies on strong assumptions and cannot always unambiguously determine the true causal structure of observational data. It therefore is necessary to carefully consider and communicate the assumptions being made by the analyst. In this light, our methodology is susceptible to being used to wrongly claim the discovery of causal structures due to careless application or intentional misuse. Particularly, the use of 'black-box' components as causal mechanisms may exacerbate concerns about identifiability, already present even for simple linear models. Whereas deep causal models can be useful for deriving insights from data, we must be cautious about their use in consequential decision-making, such as in informing policies or in the context of healthcare.

## Acknowledgements

We thank Athanasios Vlontzos for helpful comments on a draft of this paper and the anonymous reviewers for numerous constructive suggestions. This research received funding from the European Research Council (ERC) under the European Union's Horizon 2020 research and innovation programme (grant agreement No 757173, project MIRA, ERC-2017-STG). NP is supported by a Microsoft Research PhD Scholarship. DC and NP are also supported by the EPSRC Centre for Doctoral Training in High Performance Embedded and Distributed Systems (HiPEDS, grant ref EP/L016796/1). We gratefully acknowledge the support of NVIDIA with the donation of one Titan X GPU. The UK Biobank data is accessed under Application Number 12579.

## Footnotes

[2]SCMs are also known as (nonlinear) structural equation models or functional causal models.

[3]Note that here we assume full observability, i.e. no variables are missing when predicting counterfactuals. We discuss challenges of handling partial evidence in Section 6.

[4]Ventricles are fluid-filled cavities identified as the symmetric dark areas in the centre of the brain.

[5]Note that Fig. 5a shows $s$ with a unidirectional arrow from $\epsilon_S$: since $s$ has no causal parents in this SCM, abduction of $\epsilon_S$ is not necessary. If it had parents and we wished to estimate discrete counterfactuals under upstream interventions, this could be done with a Gumbel–max parametrisation as described in Appendix C.

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
