[Supplementary Material]

# Supplementary Material: Deep Structural Causal Models for Tractable Counterfactual Inference

## A Synthetic Morpho-MNIST Experiment

### A.1 Data Generation

We use the original MNIST dataset [1] together with the morphometric measurements introduced with Morpho-MNIST [2] to add functionality to measure intensity as well as set the intensity and thickness to a given value.

We implement MeasureIntensity by following the processing steps proposed by Castro et al. [2], and measure the intensity $i$ of an image as the median intensity of pixels within the extracted binary mask. Once the intensity is measured, the entire image is rescaled to match the target intensity, with values clamped between 0 and 255 (images are assumed to be in unsigned 8-bit format).

Originally, Morpho-MNIST only proposed relative thinning and thickening operations. We expand those operations to absolute values by calculating the amount of dilation or erosion based on the ratio between target thickness and measured thickness.

Finally, we follow Eq. (7) to modify each image within the MNIST dataset and randomly split the original training set into a training and validation set. We show random samples from the resulting test set in Fig. A.1.

Figure A.1: Random exemplars from the synthetically generated Morpho-MNIST test dataset

### A.2 Experimental Setup

We use (conditional) normalising flows for all variables apart from the images, which we model using (conditional) deep encoder-decoder architectures. The flows consist of components that constrain the support of the output distribution (where applicable) and components relevant for fitting the distribution. We use unit Gaussians as base distributions for all exogenous noise distributions $P(\epsilon)$ and, if available, we use the implementations in PyTorch [3] or Pyro [4] for all transformations.

22 Otherwise, we adapt the available implementations, referring to [5] for details. We indicate with $\theta$
23 the modules with learnable parameters.

24 We model the mechanisms of the thickness $t$ and intensity $i$ as

$$t := f_T(\epsilon_T) = (\exp \circ \text{AffineNormalisation} \circ \text{Spline}_\theta)(\epsilon_T)\,, \tag{A.1}$$

$$i := f_I(\epsilon_I; t) = \big(\text{AffineNormalisation} \circ \text{sigmoid} \circ \text{ConditionalAffine}_\theta(\hat{t})\big)(\epsilon_I)\,. \tag{A.2}$$

25 In the independent model, where $i$ is not conditioned on $t$, we use instead

$$i := f_I(\epsilon_I) = (\text{AffineNormalisation} \circ \text{sigmoid} \circ \text{Spline}_\theta \circ \text{Affine}_\theta)(\epsilon_I)\,. \tag{A.3}$$

26 We found that including normalisation layers help learning dynamics[1] and therefore include flows
27 to perform commonly used normalisation transformations. For doubly bounded variable $y$ we learn
28 the flows in unconstrained space and then constrain them by a sigmoid transform and rescale to the
29 original range using fixed affine transformations with bias $\min(Y)$ and scale $[\max(Y) - \min(Y)]$.
30 We constrain singly bounded values by applying an exponential transform to the unbounded values
31 and using an affine normalisation equivalent to a whitening operation in unbounded log-space. We
32 denote those fixed normalisation transforms as AffineNormalisation and use a hat to refer to the
33 unconstrained, normalised values (e.g. $\widehat{\mathbf{pa}}_k$). The $\text{Spline}_\theta$ transformation refers to first-order neural
34 spline flows [5], $\text{Affine}_\theta$ is an element-wise affine transformation, and sigmoid refers to the logistic
35 function. $\text{ConditionalAffine}_\theta(\cdot)$ is a regular affine transform whose transformation parameters are
36 predicted by a context neural network taking $\cdot$ as input. In the case of $f_I(\epsilon_I; t)$, the context network
37 is represented by a simple linear transform. Further, we model $x$ using a low-level flow:

$$h_X(u_X; \mathbf{pa}_X) = [\text{Preprocessing} \circ \text{ConditionalAffine}_\theta(\widehat{\mathbf{pa}}_X)](u_X)\,, \tag{A.4}$$

38 where the ConditionalAffine transform practically reparametrises the noise distribution into another
39 Gaussian distribution and Preprocessing describes a fixed preprocessing transformation. We follow
40 the same preprocessing as used with RealNVP [6]. The context network for the conditional affine
41 transformation is the high-level mechanism $g_X(z_X; \mathbf{pa}_X)$ and is implemented as a decoder network
42 that outputs the bias for of the affine transformation, while the log-variance is fixed to $\log \sigma^2 = -5$.
43 We implement the decoder network as a CNN:

$$
\begin{aligned}
g_X(z_X; \mathbf{pa}_X) = (&\text{Conv}_\theta(1; 1; 1; 0) \circ \text{ConvTranspose}_\theta(1; 4; 2; 1) \circ \text{ReLU} \circ \text{BN}_\theta \\
&\circ \text{ConvTranspose}_\theta(64; 4; 2; 1) \circ \text{Reshape}(64, 7, 7) \\
&\circ \text{ReLU} \circ \text{BN}_\theta \circ \text{Linear}_\theta(1024) \\
&\circ \text{ReLU} \circ \text{BN}_\theta \circ \text{Linear}_\theta(1024))([z_X, \widehat{\mathbf{pa}}_X])\,,
\end{aligned}
\tag{A.5}
$$

44 where the operators describe neural network layers as follows: BN is batch normalisation; ReLU
45 the ReLU activation function; $\text{Conv}(c; k; s; p)$ and $\text{ConvTranspose}(c; k; s; p)$ are a convolution or
46 transposed convolution using a kernel with size $k$, a stride of $s$, a padding of $p$ and outputting $c$
47 channels; $\text{Linear}(h)$ is a linear layer with $h$ output neurons; and $\text{Reshape}(\cdot)$ reshapes its inputs into
48 the given shape $\cdot$. Lastly, $[z_X, \mathbf{pa}_X]$ denotes the concatenation of $z_X$ and $\mathbf{pa}_X$, and $z_X \in \mathbb{R}^{16}$.

49 Equivalently, we implement the the encoder function as a simple CNN that outputs mean and
50 log-variance of a independent Gaussian:

$$
\begin{aligned}
e_X(x; \mathbf{pa}_X) = (&[\text{Linear}_\theta(16), \text{Linear}_\theta(16)] \circ [\text{LeakyReLU}(0.1), \widehat{\mathbf{pa}}_X] \\
&\circ \text{BN}_\theta \circ \text{Linear}_\theta(100) \circ \text{Reshape}(128 \cdot 7 \cdot 7) \\
&\circ \text{LeakyReLU}(0.1) \circ \text{BN}_\theta \circ \text{Conv}_\theta(128; 4; 2, 1) \\
&\circ \text{LeakyReLU}(0.1) \circ \text{BN}_\theta \circ \text{Conv}_\theta(64; 4; 2, 1))(x)\,,
\end{aligned}
\tag{A.6}
$$

51 where $\text{LeakyReLU}(\ell)$ is the leaky ReLU activation function with a leakiness of $\ell$.

52 We use Adam [7] for optimisation with batch size of 256 and a learning rate of $10^{-4}$ for the encoder-
53 decoder and 0.005 for the covariate flows. We set the number of particles (MC samples) for estimating
54 the ELBO to 4. We use 32 MC samples for estimating reconstruction and counterfactuals. We train
55 all models for 1000 epochs and report the results of the model with the best validation loss.

on the variable with largest magnitude. This phenomenon should be investigated further.

## A.3 Additional Results

Here we further illustrate the associative, interventional, and counterfactual capabilities of the trained independent, conditional, and full models.

### A.3.1 Association

| (a) Independent | (b) Conditional | (c) Full |

Figure A.2: Random samples generated by the independent, conditional and full model. Note how all models appear to have the same unconditional generation capacity.

| (a) Independent | (b) Conditional | (c) Full |

Figure A.3: Conditional samples generated by the independent, conditional, and full model. The high-level noise, $z_X$, is shared for all samples from each model, ensuring the same 'style' of the generated digit. The independent model generates images independent of the thickness and intensity values, resulting in identical samples. For the conditional and full models, thickness and intensity change consistently along each column and row, respectively.

Figure A.4: Reconstructions. These are computed as Monte Carlo averages approximating $\mathbb{E}_{Q(z_X|e_X(x;\mathbf{pa}_X))}[g_X(z_X;\mathbf{pa}_X)]$, where $e_X$ and $g_X$ are the image encoder and decoder networks. All models seem capable of producing faithful reconstructions.

Figure A.5: Comparison of the target covariates and the corresponding values measured from the generated images. The leftmost column refers to the accuracy of the SetThickness and SetIntensity transforms used in generating the synthetic dataset, and the remaining three columns describe the fidelity of samples generated by each of the learned models. While images sampled from the independent model are trivially inconsistent with the sampled covariates, the conditional and full models show comparable conditioning performance.

### A.3.2 Intervention

Figure A.6: Difference between conditioning and intervening, based on the trained full model. The joint density $p(t, i)$ is shown as contours in the background, for reference, and the 'violin' shapes represent the density of one variable when conditioning or intervening on three different values of the other variable. Since $t$ causes $i$, notice how $p(t \mid i)$ (left) is markedly different from $p(t \mid \mathrm{do}(i))$ (middle), which collapses to $p(t)$. On the other hand, $p(i \mid \mathrm{do}(t))$ and $p(i \mid t)$ (right) are identical.

## A.3.3 Counterfactual

Figure A.7: Original samples and counterfactuals from the full model. The first column shows the original image and true values of the non-imaging data. The even rows show the difference maps between the original image and the corresponding counterfactual image. We observe that all counterfactuals preserve the digits' identity and style. Our model even generates sensible counterfactual images (with some artefacts) in very low-density regions, e.g. '0' with $\mathrm{do}(i = 64)$ (thick but dim), and very far from the original, e.g. '2' with $\mathrm{do}(t = 5.0)$.

# B  Brain Modelling

## B.1  Data Generation

The original three-dimensional (3D) T1-weighted brain MRI scans have been pre-processed by the data providers of the UK Biobank Imaging study using the FSL neuroimaging toolkit [8]. The pre-processing involves skull removal, bias field correction, and automatic segmentation of brain structures. In addition, we have rigidly registered all scans to the standard MNI atlas space using an in-house image registration tool, which enabled us to extract anatomically corresponding mid-axial 2D slices that were used for the experiments presented in this paper. The 2D slices were normalised in intensity by mapping the minimum and maximum values inside the brain mask to the range $[0, 255]$. Background pixels outside the brain were set to zero. Age and biological sex for each subject were retrieved from the UK Biobank database along with the pre-computed brain and ventricle volumes. These volumes are derived from the 3D segmentation maps obtained with FSL, and although these

are image-derived measurements, they may serve as reasonable proxies of the true measurements within our (simplified yet plausible) causal model of the physical manifestation of the brain anatomy.

Figure B.1: Random examplars from the test set of the adopted UK Biobank dataset

## B.2 Experimental Setup

The setup for the brain imaging experiment closely follows the MNIST example as described in Appendix A.2. We randomly split the available $13,750$ brain images into train, validation and test sets with the respective ratios $70\%$, $15\%$ and $15\%$. During training, we randomly crop the brain slices from their original size of $233\,\mathrm{px} \times 197\,\mathrm{px}$ to $192\,\mathrm{px} \times 192\,\mathrm{px}$ and use center crops during validation and testing. The cropped images are downsampled by a factor of 3 to a size of $64\,\mathrm{px} \times 64\,\mathrm{px}$.

We use the same low-level mechanism for the image $x$ as with MNIST images but change the encoder and decoder functions to a deeper architecture with 5 scales consisting of 3 blocks of $(\mathrm{LeakyReLU}(0.1) \circ \mathrm{BN}_\theta \circ \mathrm{Conv}_\theta)$ each as well as a linear layer that converts to and from the latent space with 100 dimensions. We directly learn the binary probability of the sex $s$ and use the following invertible transforms to model the age $a$, brain volume $b$, and ventricle volume $v$ as

$$a := f_A(\epsilon_A) = \big(\exp \circ \mathrm{AffineNormalisation} \circ \mathrm{Spline}_\theta\big)(\epsilon_A)\,, \tag{B.1}$$

$$b := f_B(\epsilon_B; s, a) = \big(\exp \circ \mathrm{AffineNormalisation} \circ \mathrm{ConditionalAffine}_\theta([s,\widehat{a}])\big)(\epsilon_B)\,, \tag{B.2}$$

$$v := f_V(\epsilon_V; a, b) = \big(\exp \circ \mathrm{AffineNormalisation} \circ \mathrm{ConditionalAffine}_\theta([\widehat{b},\widehat{a}])\big)(\epsilon_V)\,, \tag{B.3}$$

where the context networks are implemented as a fully-connected network with 8 and 16 hidden units, and a $\mathrm{LeakyReLU}(0.1)$ nonlinearity.

## B.3 Additional Results

Likewise, we present more detailed analyses of the model trained on UK Biobank brain images and covariates, in terms of modelling the observational distribution and computing various counterfactual queries. (Continued on the next page.)

 **B.3.1 Association**

Figure B.2: Random samples from the model trained on the UK Biobank dataset

Figure B.3: Conditional samples from the model trained on the UK Biobank dataset. Images in each 3×3 block share the same the high-level noise vector, $z_X$. Each row consistently changes the brain size, whereas each column changes the ventricle volume.

Figure B.4: Original samples and reconstructions from the model trained on the UK Biobank dataset

(a) Age vs. brain volume: $p(a, b \mid s)$. Here we see differences in head size across biological sexes (reflected in brain volume), as well as a downward trend in brain volume as age progresses.

(b) Age vs. ventricle volume: $p(a, v \mid b \in \cdot)$. As expected from the literature [9], we observe a consistent increase in ventricle volume with age, in addition to a proportionality relationship with the overall brain volume.

Figure B.5: Densities for the true data (KDE) and for the learned model. The overall trends and interactions present in the true data distribution seem faithfully captured by the model.

 **B.3.2    Counterfactual**

Figure B.6: Original samples and counterfactuals from the model trained on the UK Biobank dataset. The first column shows the original image and true values of the non-imaging data. The even rows show the difference maps between the original image and the corresponding counterfactual image.

## C Discrete counterfactuals

As mentioned in the main text, the DSCM framework supports not only low- and high-dimensional continuous data, but also discrete variables. In particular, discrete mechanisms with a Gumbel–max parametrisation have been shown to lead to counterfactuals satisfying desirable properties [10]. For example, they are invariant to category permutations and are stable, such that increasing the odds only of the observed outcome cannot produce a different counterfactual outcome. More computational details and properties of the Gumbel distribution are found in Maddison and Tarlow [11].

Consider a discrete random variable over $K$ categories, $y$, with a conditional likelihood described by logits $\boldsymbol{\lambda}$, assumed to be a function $g_Y$ of its parents, $\mathbf{pa}_Y$:

$$P(y = k | \mathbf{pa}_Y) = \frac{e^{\lambda_k}}{\sum_{l=1}^{K} e^{\lambda_l}}, \qquad \boldsymbol{\lambda} = g_Y(\mathbf{pa}_Y). \qquad \text{(C.1)}$$

Under the Gumbel–max parametrisation, the mechanism generating $y$ can be described as

$$y := f_Y(\boldsymbol{\epsilon}_Y; \mathbf{pa}_Y) = \arg\max_{1 \leq l \leq K} (\epsilon_Y^l + \lambda_l), \qquad \epsilon_Y^l \sim \text{Gumbel}(0,1). \qquad \text{(C.2)}$$

Samples from the $\text{Gumbel}(0,1)$ distribution can be generated by computing $-\log(-\log U)$, where $U \sim \text{Unif}(0,1)$.

The Gumbel distribution has certain special properties [11] that enable tractable abduction. Given that we observed $y = k$, samples can be generated from the exact posterior $P(\boldsymbol{\epsilon}_Y | y = k, \mathbf{pa}_Y)$:

$$\begin{aligned}
\epsilon_Y^k &= G_k + \log \sum_l e^{\lambda_l} - \lambda_k, & G_k &\sim \text{Gumbel}(0,1), \\
\epsilon_Y^l &= -\log(e^{-G_l - \lambda_l} + e^{-\epsilon_Y^k - \lambda_k}) - \lambda_l, & G_l &\sim \text{Gumbel}(0,1), \quad \forall l \neq k.
\end{aligned} \qquad \text{(C.3)}$$

Finally, given an upstream counterfactual intervention such that $\widetilde{\boldsymbol{\lambda}} = \widetilde{g}_Y(\widetilde{\mathbf{pa}}_Y)$, the counterfactual outcome for $y$ can be determined simply as

$$y = f_Y(\boldsymbol{\epsilon}_Y; \widetilde{\mathbf{pa}}_Y) = \arg\max_{1 \leq l \leq K} (\epsilon_Y^l + \widetilde{\lambda}_l). \qquad \text{(C.4)}$$

Note that this entire derivation applies to a truly discrete variable, without the need for continuous relaxations as commonly used in deep generative models [12, 13], as the likelihood is given in closed form and no gradients of expectations are necessary.

## Footnotes

[1]We observed that not normalising the inputs can lead to the deep models prioritising learning the dependence