[Reviews · NeurIPS 2020]

Review 1

Summary and Contributions: This paper presents a framework to learn structural causal models with deep neural networks as causal mechanisms. Previous works have explored combining deep neural networks with structural causal models to estimate the effect of interventions but cannot perform counterfactual inference due to an intractable abduction step. Hence, the main novelty of this work consists of a deep SCM formulation with a tractable abduction step. The proposed method shows good performance on both real and simulated datasets.

Strengths: The problem addressed in this work is of high significance since the ability to estimate the effect of interventions and reason about counterfactual is necessary to extend the scope of machine learning beyond simple statistical association. The distinctive element of this work, a tractable abduction step, appears to be novel. Other: ------- * The presentation of counterfactuals is very clear * Assumptions and limitations are discussed (e.g., par 3 of Conclusion) * The method is evaluated on simulated and real data. Both sets of results show impressively good performance and the methodology seems mostly correct (see comment under "Correctness").

Weaknesses: (1) Soft vs hard interventions: The method is presented as supporting both hard interventions (e.g., setting thickness to 100) and soft interventions (i.e., altering the causal mechanism). However, results are only presented for hard interventions. It is thus difficult to evaluate the performance of the method in the latter case. How would one use this method to estimate soft interventions? It seems like the morpho-MNIST dataset could be used to answer this question (i.e., alter the causal mechanism for intensity). (2) "Amortized implicit" model: no experiments are reported for this model (3) Comparison to the state of the art: The authors claim that SOTA methods do not support tractable counterfactual estimation, but that some are able to estimate the effect of interventions (rung 2 of Pearl's ladder). Hence, it should be possible to compare the performance of such methods and the proposed one for estimating interventions, which was not done in the paper. Adding such a comparison would strengthen the superiority of the proposed method for all types of causal queries (association, interventions, counterfactual). Edit: ------- I thank the authors for carefully addressing my concerns. I am satisfied with their response to the concerns outlined above: (1) This was a mistake on my end and I had misinterpreted the caption of Figure 3. I consider this resolved. (2) I don't see a problem with the mention of "amortized implicit" as long as it's clarified that their evaluation is left for future work. (3) The authors make the point that their main contribution is at the level of counterfactuals and thus, their method cannot be expected to achieve top performance on levels 1-2. While I find it would have been interesting to see such a comparison, they make a valid point.

Correctness: * Section 5 -- causal model for brain imaging: The authors posit a causal graph for brain imaging and assume it's correctness in the experiments that follow. Some assumptions encoded in the graph, such that age affects brain and ventricule volume, are supported by references. However, others, like the fact that sex only affects brain volume and not ventricule volume, are not supported by references and thus, there is no way of verifying their correctness.

Clarity: Overall, the paper is very well written and the many included figures make for a very clear presentation. * Figure 1: The legend mentions that black circles indicate inputs, but some circles are white. * L175: While it is not difficult to infer the functional form of h_k from the explanation, it would help clarity to explicitly mention it. * Figure 5a: Why no double arrow for sex? I assume that this is a mistake, since counterfactuals are computed for this variable. * Discussion (L274-275): “should study more closely the dynamic behaviour of deep mechanisms in SCMs” — could the authors clarify what they mean here?

Relation to Prior Work: I believe that the Tran and Blei (2018) paper “ Implicit Causal Models for Genome-wide Association Studies” (https://openreview.net/forum?id=SyELrEeAb) is a relevant related work and that it should be discussed.

Reproducibility: Yes

Additional Feedback: It would be interesting to see how the method behaves in extrapolation, e.g., when interventions take on values beyond what was observed in the training data. For instance, in the Morpho-MNIST case study, how does the model behave when the thickness is increased to extreme values? Achieving good performance in this setting would allow to estimate the effect of previously unseen interventions. However, this might be hard to achieve without making stronger assumptions on the functional form of causal mechanisms.


Review 2

Summary and Contributions: This paper studies the evaluation of counterfactual queries using deep learning, provided with the parametrization of the underlying structural causal model (SCM). In particular, the authors focus on the Markovian causal models where exogenous variables are mutually independent. That is, these exists no unobserved confounder. The authors assume that each structural function f in SCM is a functional composition h(g(x)), where g is a convolutional neural network (NN) and h is an invertible function. The authors describe the deep learning methods of performing Pearl’s algorithm of evaluating counterfactual probabilities in such a family of parametrized SCMs, which includes a three-step procedure of abduction, action and prediction. The authors then perform simulations on synthetic datasets, showing that the proposed methods could correctly compute a special type of counterfactual probabilities, i.e., the interventional distribution induced by atomic interventions.

Strengths: This paper seems to be technically sound. Most of the claims are supported by empirical evidence. The proposed methods can perform Pearl's three-step algorithm of counterfactual evaluation in a general family of SCMs with complex mechanisms. To make the inference feasible in practical settings, the authors assume a certain parametric form of the structural functions and the exogenous distributions. The derivations behind the proposed methods seem to be reasonable. However, since I am not from a deep learning background, I cannot evaluate its novelties. Also, I appreciate the discussion in the broader impact section. The authors mention several interesting points. In particular, I second the authors' caution of drawing causal conclusions based on untestable assumptions, which could lead to careless applications and misuses.

Weaknesses: My main concern/question is that this paper considers only the Markovian causal models, where the unobserved confounders (UCs) do not exist. However, in practice, the bias due to UCs is prevalent and is arguably one of the most critical challenges of causal inference. In Markovian models, the observational distribution coincides with the interventional distribution. Therefore, a learned SCM that could perfectly mimic the observational distribution is guaranteed to estimate all interventional probabilities induced by the actual model correctly. I don't mean to take anything away from the computational challenges of learning a Markovian model with complex mechanisms. However, from the perspective of a causal inference researcher, this paper's results do not seem to be surprising, and its significance is somewhat limited. Still, it would be interesting if the author could show when the causal effect (interventional distribution) is identifiable (e.g., in Markovian models), deep learning methods could outperform other state-of-art causal estimation methods, e.g., the inverse propensity weighing. Unfortunately, these methods are not included in the experiments. With this being said, I think this paper would be most improved by further investigating causal estimation in semi-Markvoian models where UCs exist and the target effect is identifiable. Also, comparisons with other SOTA causal estimation methods are encouraged.

Correctness: The deep learning implementation for counterfactual evaluation seems to be reasonable. I have read through the experiment section. The simulation results support the authors' claims: the proposed method can estimate the interventional distribution in Markovian models.

Clarity: This paper is well written and well organized.

Relation to Prior Work: The references to related work are mostly sufficient. However, I would suggest including the following work: Louizos, Christos, et al. "Causal effect inference with deep latent-variable models." Advances in Neural Information Processing Systems. 2017.

Reproducibility: Yes

Additional Feedback: -- POST REBUTTAL -- I have read the authors’ responses and other reviewers’ comments. Unfortunately, some of my primary concerns have not been addressed, which I will elaborate on below. This paper studies the implementation of Pearl’s in a SCM, where each of its functions is represented as a neural network. The authors claim that the proposed approaches “are capable of all three levels of Pearl’s ladder of causation: association, intervention, and counterfactuals giving rise to a powerful new approach for answering causal questions in imaging applications and beyond.” However, I believe the significance of its contributions to the causal inference literature is a bit overstated. In particular, the authors assume that detailed parameterization of the target SCM is *precisely known*. Such an assumption seems to be a bit unrealistic since it does not address one of the main challenges in the causal inference: the actual model is almost always *unknown* to the learner. One may fit a structural causal model that induces the same observational distribution. However, the fitted model is not necessarily the same as the ground truth. Thus, the counterfactual probability is generally underdetermined by the observational data, even when the Markovian property is assumed (For further discussion, see (Pearl, 2000, Sec. 1.4.4)). Therefore, most causal inference literature focus on the identifiability, where the goal is to determine the target measure is uniquely discernible from the data provided with theoretical assumptions about the data-generating process; the actual model is never revealed. Since the actual causal model is often unknown in many practical settings and obtaining such a model is, in theory, challenging, applications of the proposed methods in this paper seem to be somewhat limited. Due to this reason, I would like to keep my original score. Having said that, I believe this work would be most improved by combining with the identifiability analysis of counterfactuals; the proposed methods might be applied to estimate counterfactuals when the identifiability is entailed. Applications in the causal identification with the presence of unobserved confounders are also encouraged.


Review 3

Summary and Contributions: In the context of structural causal models (SCMs) with functional relationships modeled using deep networks, this paper explores the use of normalizing (gradient) flows and/or amortized variational inference to learn the exogenous noise (potentially up to approximation) and use these to compute counterfactuals. Case studies on both synthetic and real image datasets show the value of being able to compute counterfactuals. It is claimed that all previous work in deep networks for SCMs was unable to compute counterfactuals, however this seems to rely on particular definitions of interventions and counterfactuals which I do not believe are consensus views.

Strengths: Applying advances in training neural networks to SCMs is potentially a powerful method, as the empirical experiments demonstrate. The work is highly relevant to NeurIPS.

Weaknesses: The paper suffers from a common weakness of the deep learning applications to SCMs that they do not sufficiently engage with the existing literature on causal inference. Since causality is closely connected to interacting with the real world, issues like model misspecification (e.g. unobserved confounding) are first rate concerns. If a method relies on strong assumptions, like the absence of confounding, this must be stated very clearly as a limitation. This is true about the current paper where exogenous noise variables are assumed to only directly affect one covariate. This is briefly acknowledged in the conclusion, but readers may not know of the danger for arriving at misleading results if they forge ahead with the current method in a setting where there is confounding. Regarding novelty, the paper makes claims about limitations of previous deep SCM work that aren’t fully explained. For example, the present paper claims to be the first DSCM approach that can compute counterfactuals, but one of the cited papers, the method called “causal GAN,” can create counterfactual images like “what if this person had a mustache?” Counterfactuals and interventions have (philosophically and mathematically) contested definitions, for example I do not think everyone agrees with Pearl’s ladder of causation, or that counterfactuals are necessarily retrospective, so it may be helpful to specify that a particular usage has been adopted for the current paper. Similarly, the word “mechanism” is a loaded term in causal settings, and not universally used to represent the arrows/functions in an SCM, which I have typically seen referred to as “structural equations” instead.

Correctness: Mostly yes, see my other comments.

Clarity: Yes, but see my suggestions.

Relation to Prior Work: I’ve mentioned issues here with statements about the limitations of previous DSCM work in comments the authors can see.

Reproducibility: Yes

Additional Feedback: Additional guidance about the modeling assumptions and limitations would be helpful. As it is now there are only passing comments about missing data in the discussion and identifiability in the Broader Impacts section. Related to my first comment under the “weaknesses” section, a little more explanation around Case Study 1 could potentially illustrate the dangers of misspecified models. For example, the reason the conditional model is getting the wrong distribution for intensity as shown in Figure 3 is due to the assumption of no arrow from t to i as shown in Figure 2b. I suggest spelling this out a little more. Also, if the study is expanded by including another case with unobserved confounding, for example if the true generative model has correlation between the noise variables of the intensity and thickness, this could show that even the full model could produce misleading results if an underlying assumption is violated. Related to my second comment under the “weaknesses” section, either correct the novelty claims if they are not justified in their current generality or explain why they are justified despite the counterexample I mentioned—why are the counterfactuals in the present paper different, reaching new heights on Pearl’s ladder of causation, than the counterfactuals computed in previous works? Edit after author response: since the "intervention vs counterfactual" distinction is key for the paper's novelty claim I still think this distinction needs a little more explanation, perhaps through a simple example.


Review 4

Summary and Contributions: This paper focuses on a unified framework for structural causal models using deep neural models. As for counterfactual inference, variational inference and normalising flow are employed to infer the exogenous noise. Two case studies are conducted on a synthetic dataset Morpho-MNIST and a real-world brain MRI dataset, respectively.

Strengths: [Writing and organization] The writing and organization of this paper are good. [Invertible/Amortised explicit/implicit mechanisms] This paper considers three types of mechanism based on invincibility. The discussion is comprehensive and specific. [Step-by-step counterfactual inference] The three steps, abduction, action, and prediction, are analysed in detail for counterfactual inference in implementation. [Experiments] The experiments are conducted on both synthetic and real-world datasets. The experiments are designed to evaluate the ability of both intervention and counterfactuals.

Weaknesses: [Evaluation metrics for case 1] In case 1, the lower bound and log-likelihoods are used as the evaluation metrics. Could more motivation, reasons or references behind this be provided in the rebuttal? [Evaluation for counterfactuals] The counterfactuals are evaluated via visualization. For the synthetic dataset, is it possible to evaluate counterfactuals by comparing the counterfactuals with synthetic ones? For the brain MRI scans, how can the counterfactuals be evaluated? Is the prior knowledge about the causal relations of age, sex and volumes on the image available from the dataset? --------------------------- Updated: I have read the authors' feedback. It is good to provide MAE on the synthetic dataset. I agree that it is difficult to report results like MAE on the real datasets like MRI scans. As I am not familiar with the evaluation metrics (and some readers may not either), it would be great to provide some references for Evaluation metrics for case 1. To me, it is not clear why the lower bounds are important. An intuitive explanation would be better.

Correctness: Yes.

Clarity: Yes.

Relation to Prior Work: Yes.

Reproducibility: Yes

Additional Feedback: [SCM for Morpho-MNIST] How is the SCM for Morpho-MNIST in Eq. (7) obtained? Is it pre-defined based on the dataset, i.e., the dataset is constructed based on the SCM?

[Author Response · NeurIPS 2020]

We thank the reviewers for their constructive assessment and for the many positive remarks regarding high significance and relevance (**R1**+**R3**), technical contribution (**R1**+**R2**), correctness (**R2**+**R4**), clarity (**R1**+**R2**+**R4**), and discussion of limitations and impact (**R1**+**R2**), in addition to a perceived consensus that the experiments support our claims. Thus, we find **R1** and **R2**'s lower scores surprising, given their otherwise supportive statements. We kindly invite each reviewer to revisit their recommendation in light of our rebuttal, as most concerns are easily addressed with minor clarifications.

**[R2+R3] Adopted definitions of counterfactuals etc.:** Though we attempted to distinguish our usage of 'interventions' vs. 'counterfactuals' in L53–58, we acknowledge that these definitions are not universal and may differ across causality paradigms. Following Pearl [2] and Peters et al. [1], 'interventions' operate at the population level, providing aggregate statistics about the effects of actions (i.e. noise sampled from the prior, $P(\epsilon)$). In contrast, a 'counterfactual' is a retrospective hypothetical query at the unit level (cf. potential outcome), where the structural assignments ('mechanisms') are changed but the exogenous noise is identical to that of the observed datum ($P(\epsilon|\mathbf{x})$). Further, note that an unseen combination of antecedents (e.g. 'woman+mustache' in [14]) is not a counterfactual under this definition. As suggested by **R3**, we will clarify our usage of these terms to avoid ambiguity in the final version.

**[R1+R2] Clarification of our contribution and comparison to SOTA:** **R1** and **R2** wished to see a comparison of our approach against level-2 SOTA methods in deep-learning and causal-inference literatures, respectively. However, as outlined in the title, abstract, and throughout the text, the principal claim of this manuscript is a novel approach to tractable *counterfactual inference* using deep-learning components, hence our focus on level 3 in the experimental validation. Although fulfilling the top rung of Pearl's Ladder of Causation does indeed entail generative and interventional capabilities—as demonstrated in the main text and supplement—here we make no claims of superiority of our approach on those two levels. We believe a comparison to level-2 methods is therefore beyond the scope of the current article.

**[R2+R3] Assumptions, misspecification, and unobserved confounding:** We agree that Markovianity/no-unobserved-confounders is a strong assumption and will further highlight this explicitly earlier in the paper, rather than only at the end. We shall also put more emphasis on the interpretation of our results under the light of misspecification of the simpler baseline models (**R3**). The current formulation indeed assumes causal sufficiency: every latent variable affects a single observed variable, and all relevant dependencies are captured in the DAG. Yet, to the best of our knowledge, this is the first successful attempt to build deep structural causal models (Markovian or not) that can produce high-dimensional counterfactuals. We appreciate **R2**'s suggestion regarding semi-Markovian models, and recall that our next steps involve the handling of missing variables, which is related to unobserved confounding.

**[R3] Novelty and related work:** The remark that CausalGAN [14] 'can create counterfactual images like "what if this person had a mustache?"' is inaccurate, as such a query would relate to adding a mustache to an existing (*factual*) image, not to generating novel samples. That paper focuses entirely on interventions, regarding which it unquestionably makes a strong contribution—e.g. enabling sampling from the interventional distribution $p(\text{image}|\text{do}(\text{mustache} := 1))$. However, this is not a counterfactual in the sense of a unit-specific retrospective intervention. The paper contains a single passing mention of counterfactuals (p. 8) with no corresponding experiments and assuming the exogenous noise is known—effectively avoiding to address the challenge of abduction like the other related works cited.

**[R1] Soft interventions:** Figure 3 already illustrates soft interventions on thickness of the form $\widetilde{f}_T(\cdot) = f_T(\cdot) + \tau$, with $\tau = +1$ and $-0.5$. We note that the key challenge in computing counterfactuals within Pearl's three-step procedure is *abduction*. Both the *action* (intervention) and *prediction* (final forward pass) steps are trivially computed once abduction has been performed, by simply overwriting the assignments with arbitrary functions.

**[R4] Evaluation of counterfactuals:** Following **R4**'s suggestion, we generated synthetic counterfactual images (with $\text{do}(t+2)$) and computed the pixel-level mean absolute errors: the full model achieved 17.6, while the conditional and independent models reached 41.6 and 31.8, resp. For real datasets, counterfactual evaluation is possible only in very constrained settings, as generally true counterfactuals can never be observed. For example, assuming our brain graph is correct, we may use a second MRI scan of a brain a few years later as an approximate counterfactual on age.

**Other clarifications:** **[R1] Implicit-likelihood mechanisms:** This class of mechanisms was included for completeness; we only speculate about their feasibility and make no claims on performance. This will be emphasised in the text; evaluation is left for follow-up work. **[R4] Evaluation metrics:** The log-likelihood (and lower bounds) is the training objective being optimised (see L112–113 and L135–136); it measures the level-1 fitness of a probabilistic model to the dataset, and is comparable across different models and parametrisations. **[R4] Morpho-MNIST SCM:** We will further clarify that the model in Eq. (7) was designed by us and used to generate the synthetic dataset. **[R1] Brain SCM:** Although inspired by medical evidence, our brain SCM is admittedly oversimplified and meant only as a proof of concept. We will emphasise in the text that any conclusions drawn from a model built in this framework are strictly contingent on the correctness of the assumed SCM. **[R1+R2] Suggested references:** We appreciate the pointers to Tran & Blei (2018) and Louizos et al. (2017), which we agree will strengthen our discussion. Both introduce deep level-2 methods (implicit and amortised, resp.) that explicitly deal with unobserved confounding.

[Meta-Review · NeurIPS 2020]

The reviewers agree on the whole that this work addresses an important problem and that the paper makes sound, well-supported claims. The rebuttal did a good job at clarifying the scope of their work, largely improving the scores of the reviewers. I urge the authors to carefully update the paper to address the reviewers concerns in the final version. Examples of what to improve include: - Description of the "intervention vs counterfactual" distinction. One reviewer recommends: “since it is key for the paper's novelty claim I think this distinction needs a little more explanation, perhaps through a simple example” - Engage with the existing literature on causal inference. - State clearly the assumptions and limitations of the work. I vote to accept.